# New Potential Immune Biomarkers in the Era of Precision Medicine: Lights and Shadows in Colorectal Cancer

**DOI:** 10.3390/life12081137

**Published:** 2022-07-28

**Authors:** Angela Damato, Martina Rotolo, Francesco Caputo, Eleonora Borghi, Francesco Iachetta, Carmine Pinto

**Affiliations:** Medical Oncology, Comprehensive Cancer Center, Azienda USL-IRCCS Reggio Emilia, 42122 Reggio Emilia, Italy; martina.rotolo@ausl.re.it (M.R.); francesco.caputo@ausl.re.it (F.C.); eleonora.borghi@ausl.re.it (E.B.); francesco.iachetta@ausl.re.it (F.I.); carmine.pinto@ausl.re.it (C.P.)

**Keywords:** genetic mutations, immune-biomarkers, colorectal cancer, immune checkpoint inhibitors

## Abstract

Genetic alterations in CRC have shown a negative predictive and prognostic role in specific target therapies. The onset of immunotherapy has also undergone remarkable therapeutic innovation, although limited to a small subgroup of patients, the MSI-H/dMMR, which represents only 5% of CRC. Research is moving forward to identify whether other biomarkers can predict response to ICIs, despite various limitations regarding expression and identification methods. For this purpose, TMB, LAG3, and PD-L1 expression have been retrospectively evaluated in several solid tumors establishing the rationale to design clinical trials with concurrent inhibition of LAG3 and PD-1 results in a significant advantage in PFS and OS in advanced melanoma patients. Based on these data, there are clinical trials ongoing in the CRC as well. This review aims to highlight what is already known about genetic mutations and genomic alterations in CRC, their inhibition with targeted therapies and immune checkpoints inhibitors, and new findings useful to future treatment strategies.

## 1. Introduction

The assessment of RAS and BRAF genes and mismatch repair (MMR)/microsatellite status should be evaluated as molecular panel upfront in all cases of advanced colorectal cancer (CRC) to drive patients’ selection toward biological approved treatments, although the standard cytotoxic drugs (fluoropyrimidines, oxaliplatin, and irinotecan), remain the backbone in most of the cases without specific targets [1]. However, in the early stages of CRC (I–III), the choice of adjuvant treatment is not based on molecular targets, but we know that in stage II CRC, microsatellite instability high (MSI-H)/MMR deficiency is associated with a lower recurrence rate than MSI-low/MMR proficient tumors (HR 0.65; 95% CI 0.59–0.71) [2].

In this regard, there is a clinical need to improve molecular targets, regardless of those having received strict histology-agnostic approval. 

In the last few decades, one of the most studied mechanisms involved in cancer progression concerns the immune system and how it interacts with cancer cells. This research led to revolutionary approaches with the immune checkpoint inhibitors (ICIs) by blocking the immune checkpoint proteins or their ligands to reactivate the antitumor immune response [3,4].

Plenty of biomarkers are appearing on the scientific landscape in order to identify those patients who could really benefit from immunotherapy. Among these, certainly, the most validated in CRC is microsatellite status, which occurs as MSI in 5% of sporadic CRC, while the remaining 95% are classified as stable (MSS) or MSI-low, and the efficacy of immunotherapy is yet to be defined [3].

Further biomarkers involved in immune system regulation should be identified.

Considering the role of immune checkpoints in tumor development and immune escape, the use of anti-programmed cell death protein 1 (PD-1)/PD-Ligand 1(PD-L1) monoclonal antibodies, stimulating the immune response against the tumor, has routinely become part of clinical practice [4]. 

The prognostic and/or predictive role of PD-1/PD-L1 and CTLA-4 expression was recognized in various types of cancer, such as melanoma, lung, head and neck, urothelial cancer, and others [4]. ICIs are used in these tumors in different settings, often according to different cut-off values of PD-L1 expression evaluated by immunohistochemistry (IHC) and established by different scores [5].

Most equivocal is the role of the tumor mutational burden (TMB) due to massive molecular heterogeneity resulting in dissimilar correlations between TMB levels and prognostic outcomes such as overall survival (OS) and progression-free survival (PFS) [6,7,8,9].

The attention over other immune checkpoints increased significantly in the last years, and new potential targets were identified, such as Lymphocyte-activation gene 3 (LAG3), which is highly expressed on tumor-infiltrating lymphocytes (TILs) [10]. In some tumors, such as ovarian, melanoma, non-small-cell lung cancer (NSCLC), and gastrointestinal cancers, PD-1 was usually co-expressed with LAG3 [11,12,13,14]. Simultaneous activation of the LAG3 and PD-1 pathways in TILs results in greater T-cell exhaustion, and restraining both pathways may improve T-cell activity and immune response [15]. Elevated expression of LAG3 has seen positive impacts on PFS in estrogen receptor-negative breast cancer [16]. Hald et al. [17] reported that intraepithelial-LAG3 and stromal-LAG3 were both associated with improved disease-specific survival (DSS) and OS in NSCLC. Additionally, in esophageal squamous cell carcinoma, higher LAG3 expression was positively correlated with a better OS and PFS, especially in the patients at early stage I–II [14].

In this scenario, recognizing unresponsive patients before starting the immunotherapy would help the potential development of personalized treatment and allow patients to avoid unnecessary ICIs and toxicities. Nevertheless, the prognostic significance of all these immune checkpoint molecules remains controversial.

## 2. Oncogene Driver Mutations and Therapeutic Implications in CRC

### 2.1. RAS Mutations

In CRC, the RAS gene is mutated in KRAS in up to 40% of cases, mostly in exon 2 codons 12 and 13, and in NRAS gene in approximately 3–5% of CRCs in exon 3 (codon 61) and exon 2 (codons 12 and 13) [18]. Identification of RAS status is mandatory in clinical practice due to its role as a negative predictive factor in anti-EGFR antibody response (cetuximab, panitumumab) [19,20,21]. Compared to wild-type tumors, RAS mutant patients have a worse prognosis, both in the adjuvant and metastatic settings [22,23]. Regardless of RAS status, anti-VEGF/VEGFR-2/PDGF agents such as bevacizumab, ramucirumab, aflibercept, and regorafenib demonstrated efficacy in further lines of treatment [24,25,26,27].

A dynamic process has been identified regarding further KRAS mutations leaving the opportunity to develop new treatment strategies. For example, the KRAS G12C codon mutation, which represents 3% of all CRC cases, is now susceptible to inhibition by small molecule inhibitors that specifically and irreversibly block KRAS G12C by locking it in an inactive GDP-bound site [28,29]. Two basket phase I/II studies proved that sotorasib (CodeBreaK100) and adagrasib (KRYSTAL-1) are both active in patients with advanced solid tumors harboring a KRASG12C mutation, among which mCRC [30,31,32]. However, EGFR signaling mutations were identified as the dominant mechanism of CRC resistance to KRAS G12C inhibitors [33]. Indeed, the combination of adagrasib and cetuximab improved ORR to 43% and DCR to 100% [34], leading to a possible novel histology-tuned targeted treatment for mCRC, also being tested as sotorasib plus panitumumab [35]. A randomized phase III trial (KRYSTAL-10) [36] is ongoing and evaluates second-line agadrasib plus cetuximab versus standard chemotherapy. 

Additionally, several other combinations of sotorasib or adagrasib with diverse compounds as well as newer anti-RAS strategies are ongoing.

### 2.2. BRAF Mutations

The second genetic mutation is represented by BRAF V600E, accounting for 8–10% of mCRC cases, and is nearly always mutually exclusive with KRAS, resulting in RAS-independent oncogenic signaling through the mitogen-activated protein kinase (MAPK) pathway (RAS-RAF-MEK-ERK) [37]. This alteration is associated with a worse prognosis and could predict a poorer response to anti-EGFR treatment [38]. Based on the breathtaking results achieved in other solid tumors such as melanoma, several trials explored the use of BRAF inhibitors as single agents or in combination with MEK inhibitors in the mCRC. However, the inhibition in the MAPK/ERK pathway with a BRAF inhibitor results in adaptive feedback reactivation of MAPK signaling mediated by rapid feedback EGFR reactivation [39,40]. In order to overcome primary resistance, anti-EGFR monoclonal antibodies were tested with BRAF V600E inhibitors, with modest activity for this combination [41,42]. Recently, the BEACON phase III trial compared the combination of anti-BRAF V600E (encorafenib), anti-MEK (binimetinib), and anti-EGFR (cetuximab) inhibitors as a triplet or doublet (encorafenib and cetuximab) scheme versus investigator’s choice (FOLFIRI or irinotecan and cetuximab) in pre-treated BRAF V600E mutant mCRC patients [43]. Long-term follow-up updates showed that both double and triple combinations achieved a median OS of 9.3 months, compared with 5.9 months in the control arm. The overall response rate (ORR) was 26.8% with the triplet, 19.5% with the doublet, and 1.8% in the control arm, with prolonged maintenance of the quality of life (QoL) and reducing the risk of QoL deterioration by more than 40% [43]. The phase II ANCHOR-CRC trial is testing the triplet regimen (encorafenib, binimetinib, and cetuximab) in the first-line setting [44], showing an ORR of 47.8% and DCR of 88%; median PFS was 5.8 months (95% CI, 4.6–6.4) and mOS was 17.2 months (95% CI, 14.1-NE). 

Furthermore, approximately 20% of BRAFV600E mCRC have high-level MSI (MSI-H). Currently, an anti-PD-1 antibody, pembrolizumab, represents the standard of care for BRAFV600E mutant MSI-H mCRC [45]. In a phase I/II clinical trial, patients with treatment-refractory BRAFV600E MSS mCRC received encorafenib, cetuximab, and nivolumab, showing effective activity in terms of ORR (45%), DCR (95%), mPFS (7.3 months; 95% CI, 5.5-NA), and mOS of 11.4 months (95% CI, 7.6-NA) [46]. 

Although the best treatment has not yet been identified, an aggressive strategy involving triplet chemotherapy and targeted therapy is currently the standard of care for fit patients. BRAF- therapies combined with other anti-EGFRs, MEK inhibitors, or PI3K inhibitors seem promising [47,48].

### 2.3. ERBB2 Alterations

Further developments have identified another molecular biomarker in mCRC, as the prevalence of receptor tyrosine-protein kinase erbB-2 (ERBB2) or frequently called human epidermal growth factor receptor 2 (HER2) amplification, in approximately 3–8%, especially in KRAS wild-type CRC [49]. Although retrospective data have demonstrated that ERBB2 amplification represents a negative predictive biomarker for anti-EGFR therapies, prospective data are needed to define the relationship between ERBB2 and anti-EGFR responsiveness [50,51]. Indeed, RAS/BRAF wild-type mCRC should be screened for ERBB2 amplification before treatment with anti-EGFR therapies [52]. The activity of an anti- ERBB2-targeted therapy was demonstrated in the phase II HERACLES-A trial [53], in which a double signaling blockade by trastuzumab and lapatinib achieved an ORR of 30% in KRAS wild-type chemo refractory tumors. ERBB2 positivity was defined as tumors with a 3 + score in more than 50% of cells by immunohistochemistry or a 2 + score and a HER2:CEP17 ratio > 2 in more than 50% of cells by FISH. Similar results were confirmed in the MyPathway phase II basket trial (trastuzumab plus pertuzumab) [54], and HER2 positivity was defined as amplification (FISH or HER2/CEP17 ratio 2.0 or copy number > 6), overexpression (IHC 3 +), or activating HER2 mutations. Emergent ERBB2 target agents are evolving in mCRC, such as the tyrosine kinase inhibitors, tucatinib, and an antibody-drug conjugate, trastuzumab-deruxtecan, that in the phase II DESTINY-CRC01 trial [55] achieved in the ERBB2 positive RAS wild-type mCRC an ORR of 45.3%. The preliminary results of the ongoing phase II MOUNTAINEER trial [56] showed 55% ORR with tucatinib combined with trastuzumab.

Remains to be clarified the heterogeneity in HER2/ERBB2 assessment in clinical trials and even across different tumor types [49].

## 3. Long-Standing and Emerging Immune Biomarkers in CRC

### 3.1. Microsatellite Instability/Mismatch Repair Status

Repeated non-coding DNA sequences (microsatellites) are frequent sites for mutations during DNA replication. The mismatch repair (MMR) system works by detecting and correcting these mutations, but it can be deregulated in several types of tumors [57]. In CRC, because of genetic or epigenetic alterations causing the inactivation of MMR genes (MLH1, MSH2, MSH6, and PMS2), the MMR system is defective in about 15% of all CRCs and 4% of mCRC, resulting in microsatellite instability (MSI) and can be used as a biomarker [57]. This phenotype frequently involves the proximal colon, poorly differentiated, and mucinous histology, characterized by an accumulation of mutations, which generate frame-shifted neoantigens with great immunogenic potential [58]. MSI-H/dMMR tumors are highly infiltrated with immune cells, including CD4+ and CD8+ TILs, Th1 (T helper 1), and macrophages [57,58,59,60]. In early-stage II CRC, MSI tumors display a lower risk of recurrence (HR 0.65; 95% CI: 0.59–0.71) [2], while in stage III disease, data are conflicting [61,62].

In the metastatic setting, MSI status has been demonstrated to be a biomarker of response to immunotherapy due to an accumulation of mutation-associated neoantigens that can be recognized by the immune system. This provided the scientific rationale for using immunotherapy in this setting [59,60]. 

Several phase II/III clinical trials tested anti-PD-1/PD-L1 agents in solid tumors, including CRC [45,63,64,65,66]. Across studies, pembrolizumab showed an ORR of 40% and an advantage in mPFS and mOS in the MSI-H/dMMR population. Furthermore, anti-PD1 nivolumab demonstrated an ORR of 31%, mPFS of 14.3 months, and 12-month PFS of 73% [67]. Combining nivolumab with anti-CTLA4 antibody, ipilimumab, there were higher responses but an increase in immune-related toxicities [68]. Based on these exciting results in 2017 and 2018, first pembrolizumab and then nivolumab alone or nivolumab/ipilimumab granted accelerated approval for patients with MSI-H/dMMR CRC that had progressed to fluoropyrimidines, oxaliplatin, and irinotecan. Outstanding results come from the phase III KEYNOTE-177 clinical trial affording statistically significant improvements in terms of PFS for pembrolizumab versus chemotherapy (16.5 vs. 8.2 months; HR 0.60; *p* = 0.0002) in the first-line setting of dMMR/MSI-H mCRC patients [45]. Another humanized anti-PD1 monoclonal agent, dostarlimab, was evaluated in clinical studies. According to the last update of cohort 7 from the GARNET trial [69], MSI mCRC patients achieved 36.2% of ORR, and at data cut-off, median DOR was not reached [67]. More recently, Cercek et al. [70] showed compelling results about the use of dostarlimab in 12 locally advanced rectal cancer patients obtaining 100% of complete clinical response (95% CI 74–100) with no evidence of tumor on imaging and endoscopic evaluation, or biopsy. At the time of this report, no patients had received chemoradiotherapy or undergone surgery, and no cases of progression or recurrence are reported during follow-up up to range of 25 months. 

Despite this therapeutic success of immune therapies, drug resistance remains the most important obstacle to the achievement of durable outcomes since tumor cells evade the immune attack through PD-L1 expression in the cell membrane to block the PD-1/PD-L1 axis. Especially the destruction of the antigen-presenting complex and neoantigen defeat, along with driver alteration, could be the major causes of immune-escape in MSI tumors [71,72]. 

### 3.2. Tumor Mutational Burden

Tumor mutational burden (TMB) represents the total number of non-synonymous mutations in cancer cells [73]. It is also definite as the total of coding errors of somatic genes, deletions, insertions, or base substitutions detected across per million bases [74]. This biomarker could be used as an important prognostic factor to predict the response to immunotherapies independent of MSI status and PD-L1 expression [75,76]. Indeed, important studies hypnotized that high levels of TMB are correlated with a high density of neoantigen-specific, more tumor infiltration lymphocytes, which can lead to upregulation od PD-L1 on cancer cells [77]. High levels of mutations increase neoantigens’ burden, making tumors more immunogenic and responsive to immunotherapy [78].

Different studies analyzed the correlation between high expressions of this factor and PFS, ORR, and OS. The important role of this predictive biomarker for immunotherapy is confirmed in patients with NSCLC and melanoma, but it has not yet been confirmed in patients with CRC [79,80,81,82]. 

Tumor mutational burden is usually analyzed by two different methods: whole exome sequencing (WES) and, more often, next-generation sequencies (NGS) based on the comprehensive genomic profiles (CGP) [83,84] expressed as a ratio between the mutation number and megabases of genomic region sequenced. 

Schrock et al. underline the TMB’s predictive role in response to ICIs in MSI/dMMR mCRC [78]. To define the optimal cut-off TMB for outcome prediction, the authors considered the cut point as a lower 35th TMB-percentile of MSI CRC associated with less benefit from PDL1/PD1 inhibitors. In this study, 22 CRC patients were treated with PD1/PD-L1 inhibitors, and the response was assessed using RECIST 1.1 Criteria. Comparing responders [defined as complete response (CR)/partial response (PR)] and non-responders [stable disease (SD)/progression disease (PD)], the TMB showed a significant association with ORR (*p* < 0.0003) and PFS (*p* < 0.01). Non-responders had a median TMB of 29 mutations/Mb compared to responders with 54 mutations/Mb [78].

The TRIBE2 phase III study evaluated TMB and mismatch repair status as potential prognostic biomarkers. The TMB was evaluable in 224 cases and categorized as high (≥17 mut/Mb) in 11 patients, intermediate (7–16 mut/Mb) in 157 patients, and low (<7 mut/Mb) in 56 patients. Eight percent of MSI tumors showed a higher TMB compared to MSS tumors (39 vs. 9 mut/Mb, *p* < 0.0001). The most frequent mutated genes in tumors with different TMB expressions were ASXL1, MSH6, and ARID1A, usually mutated in high-TMB tumors; instead, TP53 and APC recurred in low/intermediate-TMB. Overall, patients with high-TMB tumors obtained a better outcome than low/intermediate ones (mPFS 17.3 vs. 10.6 months, respectively) [85].

The potential predictive role of TMB in mCRC was also investigated in a subgroup of MSS CRC patients associated with O6-Methylguanine-DNA-methyltransferase (MGMT)-deficiency, showing as using temozolomide led to the appearance of neo mutations and tumor neoantigens, prompting to immunotherapy effectiveness [86,87]. 

High levels of TMB are necessary but not sufficient to obtain benefits from immunotherapies. Patients affected by CRC with high level of TMB could not respond to treatment; however, low TMB in CRC are sometimes associated with higher activation of the WNT pathway, and this can lead to immune cold tumors [88].

These studies highlight how the TMB could be used as an important biomarker to assess the likelihood of response to immunotherapy in patients affected by CRC, nevertheless considering the lacking evidence and limitations associated with defining an optimal cut-off [83]. 

### 3.3. CD279 (PD-1)/CD274 (PD-L1) Expression

CD279 (PD-1) is a checkpoint protein expressed on activated T cells and constrains T cells activation and cytokine production by binding two ligands, CD274, also commonly referred to as PD-L1 (constitutively expressed on hematopoietic and non-hematopoietic cells) and PDCD1LG2 or PD-L2 (expressed on dendritic cells, macrophages, and mast cells) [89]. After antigen recognition, an activated T-cell expresses CD279 on its membrane, producing interferons that induce CD274 expression in various organs and tissues. In solid tumors, the interaction between CD274, present on the tumor front, and CD279 prevents T-cells clonal expansion, evading the suppressive innate immune response. Therefore, overexpression of the CD274 molecule in the tumor microenvironment compromises immune response. Several mechanisms are involved in CD274 expression improvement on tumor cells, such as the activation of intracellular signaling pathways (MAPK and PI3K-Akt), the enhanced activity of transcription factors (STAT3), and a rising occurrence of inflammatory mediators (interferon-γ and interleukin-6) [90].

In CRC, the correlation between CD274 expression, its prognostic or predictive role, and some clinical and pathological features remains unclear.

Several studies and meta-analyses have explored this issue but with controversial results. Some studies reported a low rate of CD274 expression in this tumor, ranging from 9% to 12% [91,92]. In the first study [89], the authors evaluated PD-L1 expression on tumor cells using a grading system as follows: 0: <5% of tumor cells; 1: 5–49%; 2: ≥50%. They examined only the membranous staining, not the cytoplasmic one, considering PD-L1 positive 1 and 2 scores. 

Theoretically, a high PD-L1 expression should be related to a better response to immunotherapy. However, some promising results have also been found in PD-L1 negative cancers [93], and, on the contrary, no results have been reported in high CD274 CRC. Possible reasoning could rely on different PD-L1 expressions between primary tumor and metastatic site, as reported by Wang et al. [94]. In this case, CD274 staining intensity was evaluated by another scoring system: 0 = negative, 1 = weak, 2 = moderate, and 3 = strong.

In Bae et al.’s study [95], PD-L1 expression was evaluated in patients who underwent surgery for CRC. Authors considered a low expression if ≤50% of cells were positive and high expression when >50%. The outcomes (OS, DFS, and systemic recurrence rate) were significantly better in patients with higher expression than those with low CD274 expression (OS 48.2 vs. 32.9%, *p* = 0.047; DFS 43.3 vs. 32.9%, *p* = 0.021; systemic recurrence rate 42.7 vs. 12.9%, *p* = 0.030). 

Of note, the main differences were revealed in patients with stage III CRC. Droeser et al. [96] highlighted that CD274 expression was associated with better survival in pMMR CRC (mOS 32 vs. 23 months when high vs. low or absent CD274 expression). In this early setting, another study showed completely different results, with poor prognosis in patients with CD279 and CD274 positive expression CRC [97]. In this study, even using a similar staining intensity grade as Wang et al. [91] (0+, 1+, 2+, 3+), the authors analyzed CD274 expression by dividing the percentage of CD274-positive cells into quartiles: 0+ = 0–5%; 1+ = 6–25%; 2+ = 26–50%; 3+ = 51–75%; 4+ = 76–100%. Only 3+ and 4+ scores were defined as CD274 positive expression. In the stage II CRC experience of Eriksen et al. [98], a high PD-L1 expression was significantly associated with MSI tumors but not with survival.

Further meta-analyses agreed with a poor prognostic role for CD274 in CRC [96,97,98,99,100], even with different sample sizes or high heterogeneity among them. Mainly, in the meta-analysis of ten studies by Shen et al. [100], the CD274 expression correlated with lymphatic invasion and advanced stage, but not with MSI status. Wang et al. [94] showed that PD-L1 expression was correlated with lymphatic invasion and tumor diameter but negatively with differentiation and vascular invasion.

All these results, even conflicting, suggest that a CD274 positive expression could be a negative prognostic factor in CRC, potentially helping clinicians to stratify patients for immunotherapy, especially MSI-H/dMMR patients [101,102]. In this context, evaluating CD274 in neoplastic cells and tumor-infiltrating immune cells could be helpful. Indeed, Valentini et al. [103] showed how this approach is very informative, and by combining the expression of CD274 on these two types of cells with inflammatory cells (CD274 positivity was considered as an expression on ≥5% of membranous positive cell staining), they defined three CRC groups with different clinical and pathological features and associations with MSI status. The authors suggested that the group with high CD274 expression on tumor and immune cells and with high levels of tumor-infiltrating cells could benefit from immunotherapy, compared to CD274 negative and with low immune cells CRC.

Surely, one of the significant problems related to CD274 evaluation in CRC is the assessment methodology because of different assays, scoring methods, and CD274 positivity cut-offs. As mentioned above, some studies used a cut-off value of 1%, 5%, or 50%, and different staining intensity grades [4,91,104]. Further studies evaluated CD274 in both neoplastic and immune cells, while others only in neoplastic cells. Similarly, in some studies, only the membranous staining was evaluated, while in others, both the membranous and cytoplasmic staining.

Although the expression of CD274 correlates with the dMMR CRC in several studies, its prognostic and predictive role remains unclear in these patients. Moreover, the protective effect associated with microsatellite instability and OS diverged with the poor prognostic role of CD274, as shown in some meta-analyses. In the study of Lee et al. [105], the authors confirmed that dMMR correlated significantly with CD279 and CD274 expression but also showed the potential role of CD274 expression in those dMMR CRC with a poor prognosis despite a high-level tumor-infiltrating lymphocyte. In particular, the positive prognostic impact of tumor lymphocytes in patients with dMMR CRC was denied by the presence of high-level CD274 in the tumors. In this contest, where the MSI is a strong biomarker of response to immunotherapy, the expression of CD274 could potentially help clinicians to stratify patients further for therapy with checkpoint inhibitors [106]. 

However, in the Checkmate 142 study [67,106], no statistically significant difference in survival was observed based on the level of CD274 expression (low <1% versus high ≥1%, respectively) and the presence of Lynch syndrome.

Previous studies have not clarified the difference between sporadic MSI and Lynch syndrome-associated cancers in terms of the expression of CD274. A retrospective Japanese study of CRC suggests that sporadic MSI cancers compared to Lynch-syndrome-associated cancers more frequently develop as poorly differentiated, solid-type tumors with a medullary morphology, and many express CD274 (25.0 vs. 3.6%, *p* = 0.034) [107].

The CD274 expression and Lynch syndrome status were also analyzed in the study of MSI-H/dMMR mCRC patients treated with the combination of nivolumab and ipilimumab, but no correlation between clinical response and these factors was identified. However, there was a higher ORR among patients with Lynch syndrome compared with the rest of the cohort (71% versus 48%, respectively) [106].

The predictive role of CD274 in dMMR CRC and its possible correlation with Lynch syndrome remain interesting suggestions that deserve to be explored in future studies.

### 3.4. Lymphocyte-Activation Gene 3 (LAG3)

LAG3 (lymphocyte activation gene 3) is a transmembrane receptor-function protein consisting of 498 amino acids whose genes are mapped on chromosome 12 adjacent to the CD4 gene. It belongs to the immunoglobulin superfamily and, like CD4, its extracellular region is composed of four IgG domains (D1–D4). Both LAG3 and CD4 bind class II MHC, but LAG3 possesses an “extra loop” in the D1 domain of 30 amino acids that allows stronger affinity binding, unlike CD4, which uses more amino acid residues. D1/MHC class II binding is mediated by the D2 domain [108].

The intracellular portion of LAG3 is not yet well known and contains a unique region known as KIEELE, which is necessary for the inhibitory function of T lymphocytes through a possible blockage of the S phase of the cell cycle and, subsequently, cell division (Figure 1). However, the exact mechanisms by which LAG3 inhibits T cell function are not completely known, and the intracellular ligands of KIEELE have not yet been identified [109,110,111,112]. 

In addition, another inhibitory function on CD4 T cells seems to be realized by blocking IL-2 production, which occurs by a calcium-calmodulin-dependent signal transduction mechanism that in turn activates calcineurin (CaN), which is critical for de-phosphorylation and, thus, activation of the transcription factor NFAT that cooperates in gene expression of IL-2 transcription factors (Figure 2). Again, the ways in which LAG3 exerts its inhibitory function on CD4 are not known [112].

Indeed, LAG3 is expressed on activated T cells, T-regulatory cells, NK cells, B cells, and plasmacytoid dendritic cells, where it regulates their proliferation, activation, and homeostasis. All these cell types participate in the establishment of the tumor-associated immune (TIL) microenvironment in which overexpression of LAG3 has been observed, so it could be a valuable target of immunological therapy in cancer patients [113]. In melanoma and NSCLC tumors, the blockade of LAG3 exerts antitumor effects in patients with positive LAG3 expression, and several studies are ongoing for other solid tumors [114].

In addition, the synergistic interaction of LAG3 with CD274 in the mechanisms of escape from the immune system by cancer in which both are widely expressed has been demonstrated so that a combined LAG3/CD279 blockade by antibodies has been shown to be effective in tumor-bearing mice unresponsive to the single antibody [115].

The expression of LAG3 has been assessed by IHC in most studies. Quantitative analysis of immune checkpoints has been obtained by various methods and scoring systems. It is possible to use a weighted histoscore calculated as: (% unstained tumor cells × 0) + (% weakly stained tumor cells × 1) + (% moderately stained tumor cells × 2) + (% strongly stained tumor cells × 3) to give a range from 0 to 300 [116]. 

Immune cells are evaluated using cytoplasmic and/or membrane staining by performing a positive lymphocyte count for each cell nucleus. The difference between high and low expression can be determined by ROC curves [116].

Another tool is Cybertsort, a software that provides the estimation of immune cell infiltration fraction using gene expression profiling [117]. A quantitative analysis is initially performed where staining scores are defined as 0 (negative), 1 (weak), 2 (moderate), and 3 (strong). The scores for the percentage of tumor cells were classified as 0, 1, 2, 3, and 4, respectively. The staining intensity scores were multiplied by the scores for the percentage of stained cells.

Yoshihara et al. [118] establish the ESTIMATE (Estimation of STromal and Immune cells in MAlignant Tumors using Expression data) algorithm allows the degree of infiltration of immune and stromal cells in terms of tumor purity to be assessed through their gene expression and by inferring the fraction of stromal and immune cells in tumor samples. 

A retrospective analysis was conducted to better identify the prognostic role of LAG3 [116]. In the 773 stages, I–III CRC patients underwent tumor resection surgery [106], and the expression of immune checkpoints at the stromal and tumor levels was evaluated by combining immune checkpoint stromal score (CICSS): CICSS 3 (TIM3- LAG3 and PD1 highly expressed), CICSS 2 (only two markers highly expressed), CICSS 1 (one high marker or weakly expressed markers). This study demonstrated a prognostic value of higher expression of TIM-3, PDL1- LAG3, and PD1 at the tumor associated with poor survival, while high expression at the stroma correlated with improvement of 10 years of survival (Table 1).

Conversely, Zhou et al. [119] carried out in 73 stage IV CRC patients the different expressions of lymphocyte subpopulations in the primary tumor and secondary cancer sites (peritoneal and liver lesions) and observed a higher expression of CD8+ and lower expression of CD4+ in the secondary site than the primary tumor. Consequently, CD8+ lymphocytes at liver metastasis showed a high frequency of PD-L1+, LAG 3, and TIM-3 cells compared to the primary tumor and peritoneal metastasis. In addition, CD4 fox p3+ regulatory lymphocytes in liver metastasis contained higher frequencies of PD-L1 and LAG3 than in the primary tumor and peritoneal metastasis. The expression of regulatory T lymphocytes was similar in the primary tumor, liver metastasis, and peritoneal metastasis. Thus, the higher expression of PD-L1 in liver metastases compared with primary colon cancer could make metastases more sensitive to treatment with immune checkpoint inhibitors. Finally, this study evaluated the expression of immune checkpoints in lymphocytes infiltrating CRC liver metastases by comparing a stable mismatch repair system gene with an unstable set-up. In MSS mCRC, there is an increased expression of PD1, TIM3, CTLA4, and/or LAG-3 in liver metastases compared with primary tumors, and LAG-3, TIM-3, and/or PD-1 in liver metastases compared to peritoneal metastases. In MSI mCRC, there is higher CD8+ infiltration and up-regulation of PD-L1/PD1/CTLA4/LAG3 expression in primary tumors, leading these tumors to be more responsive to immunotherapies. Interestingly, although in a small sample, the association between LAG-3 expression and PFS after liver metastases resection was also investigated. It was observed that higher expression of LAG3 at the level of the tumor-associated lymphocyte subpopulation correlates with higher PFS as an independent predictor factor.

## 4. Synergistic Immunotherapy Combinations in CRC

Preclinical data suggest that concurrent activation of the LAG3 and CD279 signal in TILs results in more T-cell exhaustion than either pathway alone, and dual inhibition of these pathways may improve T-cell function and increase immune response [120,121]. 

Based on this rationale, the synergistic combination was investigated in other solid tumors. Indeed, the phase II/III RELATIVITY-047 clinical trial evaluated in the first-line setting in patients affected by metastatic or unresectable melanoma, a combination with an anti–LAG3 antibody, relatlimab, and nivolumab, demonstrated a superior PFS for the combination compared nivolumab as a single agent (10.1 vs. 4.6 months, HR 0.75; 95% CI 0.62–0.92; *p* = 0.006 by the log-rank test), regardless of LAG3 expression [122]. 

Unfortunately, we do not yet have results in other tumors, but several trials are ongoing, especially in CRC in different stages of recruiting, also with anti-PD-1/PD-L1 and/or anti-LAG3 antibodies as single agents or in combination (Table 2). 

## 5. Conclusions

Despite notable improvements achieved in recent years with targeted therapies and immune checkpoint inhibitors in CRC, few patients benefit and unfortunately manifest early resistance. Several immune biomarkers have been described in solid tumors, including CRC, but none, except for MSI-H/dMMR, has really demonstrated a predictive impact on response to ICIs. Further investigations are necessary; hence, there is a need to promote the enrolment of mCRC patients in clinical trials to better understand immune system and tumor landscape interaction.

## Figures and Tables

**Figure 1 life-12-01137-f001:**
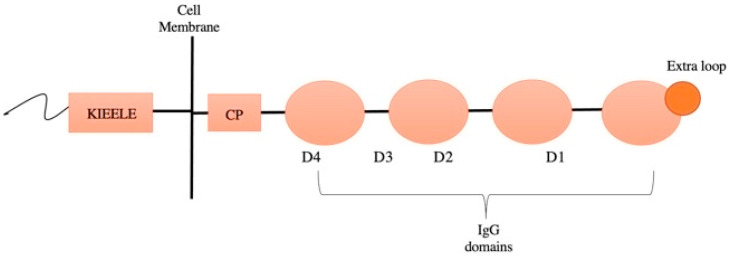
Extracellular and intracellular domains of LAG3.

**Figure 2 life-12-01137-f002:**
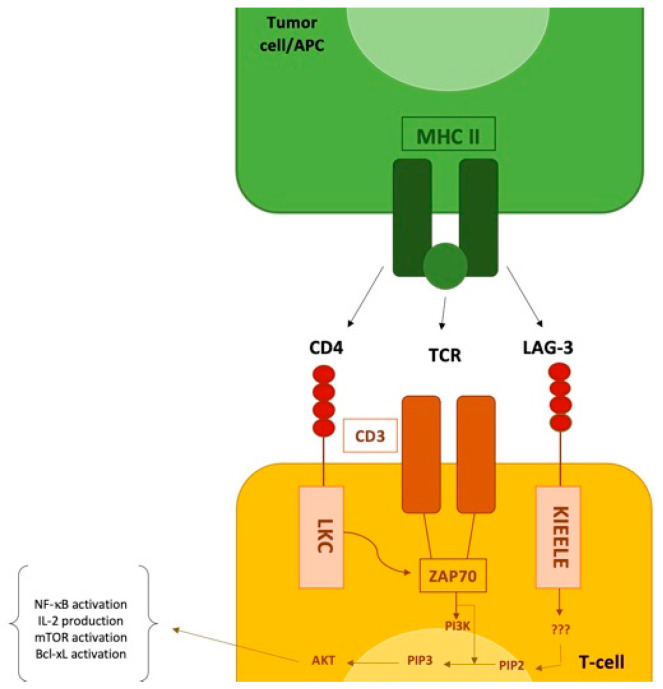
LAG3 signaling and interaction with other immune checkpoints. The interaction of LAG3 with MHC-II prohibits the binding of the same MHC molecule to a TCR and CD4, thus suppressing TCR signal. LAG-3 transmits an inhibitory signal via the KIEELE motif in the cytoplasmic tail.

**Table 1 life-12-01137-t001:** Correlation between LAG-3 expression in the tumor and stroma and 10 years of survival.

	LAG-3 on Tumor Cells		LAG-3 on Stromal Immune Cells	
	N°	Survival	N°	Survival
Low	253	73%	196	63%
High	160	65%	191	78%

**Table 2 life-12-01137-t002:** Clinical trials being recruited in CRC using anti-LAG3 and/or anti-PD-1/PD-L1 antibodies.

Trial	Phases	Setting/Line	Drugs	Biomarker	End Points
NCT03642067 [123]	II	Stage IV ≥2	Nivolumab Relatimab	MSS PD-L1/Mucin (CPM) score ≥ 15% or <15%	Primary: ORR Secondary: AEs
NCT05064059 [124]	III	Stage IV ≥2	Favezelimab/ pembrolizumab vs. Regorafenib TAS 102	MSI-H/dMMR	Primary: OS Secondary: PFS, ORR, DoR, AEs, TTD
NCT05310643 [125]	II	Stage IV 1 and ≥2	Nivolumab Ipilimumab	MSI-H/dMMR	Primary: ORR Secondary: AEs, DCR, PFS, OS, ctDNA
NCT05371197 [126]	II	Stage III Neoadjuvant	Envafolimab	MSI-H/dMMR	Primary: pCR Secondary: DFS, OS, drug safety and feasibility
NCT05118724 [127]	II	Stage III (ineligible for oxaliplatin) Adjuvant	Atezolizumab ± IMM-101	MSI-H/dMMR	Primary: 3 y DFS Secondary: 1, 2, 5 y DFS and OS
NCT04895722 [128]	II	Stage IV 1 and ≥2	Pembrolizumab Quavonlimab Favezelimab Vibostolimab MK-4830	MSI-H/dMMR	Primary: ORR Secondary: DoR, PFS, OS, AEs

CPM—combined PD-L1/mucine score; ORR—overall response rate; AEs—adverse event; OS—overall survival; PFS—progression-free survival, DoR—duration of response; TTD—time to deterioration; ctDNA—circulating tumor DNA; DFS—disease-free survival; y—years.

## Data Availability

Not applicable.

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
