# Peer review of "New Potential Immune Biomarkers in the Era of Precision Medicine: Lights and Shadows in Colorectal Cancer"

_life, 2022, doi:10.3390/life12081137_

Round 1

Reviewer 1 Report

The current review article aimed to review what is already known about genetic mutations and genomic alterations in CRC, their inhibition with targeted therapies and immune checkpoints inhibitors, and new findings useful to future treatment strategies. However, there are some significant issues (according to the fact that this review was supposed to be a conclusive narrative review) regarding writing and discussing issues. 

1- There is a narrative review by Xie et al. ( PMC7082344), which is not cited in your manuscript. The mentioned publication describes all of the issues you have mentioned and you should explain what your review is going to add to that publication.

2- The current manuscript lacks informative figures and tables. The role of each mutation could have been better described in figures.

3- The information about synonyms and abbreviations is not listed. 

4- It could be a better review if the authors described the underlying mechanisms and pathways, each mutation or a molecule could be involved and the subheadings would be according to that classfifcation.

5- A deeper literature review and search might be needed.

Thus, I would like to suggest the current submission, not suitable for publication in LIFE. 

Author Response

Dear reviewer, in the meantime, thank you for the quick and interesting review done for our work. However, there are some points where I disagree regarding the your comments:

1) About the review of Xie et al, this paper aimed to describe in detail the signaling pathways here EGFR/VEGF/HGF and related target therapies leaving less focus on immunobiomarkers and which is exactly the purpose of our review, while still giving hints to the genetic compound (in my opinion must be included since the starting point to then get to the immune-pathogenesis).
2) Regarding the figures, as mentioned above, the purpose being directed more toward new immune-biomarkers; therefore, we took the suggestion and inserted two figures about LAG3.
3) We have included the abbreviations paragraph as requested.
4) This review is not intended to describe the mechanisms and signaling pathways of each gene but to give additional information to what has already been well described in other previous papers (e.g., the review by Xie et al.) regarding new albeit, as yet unvalidated perspectives, in "immunotherapy era of CRC".
5) Further research has been done and more information and references have been included in the text.

I hope you can reconsider our work suitable for publication as something that adds to the existing literature by describing new immune-biomarkers whose role is still being studied and controversial but at the same time very fascinating to researchers.

Best regards,

Angela Damato, MD Phd

Reviewer 2 Report

The authors well summarize the highlight that already known about genetic mutations and genomic alterations in CRC, their inhibition with targeted therapies, immune checkpoints inhibitors, and new findings useful to future treatment strategies. I have a suggestion that authors can consider to further strengthen the manuscript.  

The authors used many gene names, but the authors used colloquial names. HGNC(https://www.genenames.org/) defines the official gene symbol and Fujiyoshi et al. (Opinion: Standardizing gene product nomenclature-a call to action. Proc Natl Acad Sci USA 2021. 19;118(3): e2025207118. doi: 10.1073/pnas.2025207118.) mentioned the how to mention protein name using official gene names. Based on the rule, I would suggest that the authors should use official gene symbols in order to avoid confusion.

For example, “PD-L1” should be mentioned as “CD274”, “PD-1” should be mentioned “PDCD1”, “HER2” should be mentioned as ”ERBB2”, and “LAG-3” should be mentioned as “LAG3”.

Author Response

Dear reviewer, in the mean time, thank you for your quickly ed interesting suggestion.
As you suggested, although it is much more straightforward and easier for clinicians to read PD-L1 or HER-2, I have changed those acronyms in the text to official gene symbols/name.

Best regards,
Angela Damato, MD PhD

Round 2

Reviewer 1 Report

The revised version looks fine and I was satisfied by your explanations.